# Dose–Response Effects of Bamboo Leaves on Rumen Methane Production, Fermentation Characteristics, and Microbial Abundance In Vitro

**DOI:** 10.3390/ani12172222

**Published:** 2022-08-29

**Authors:** Seong Uk Jo, Shin Ja Lee, Hyun Sang Kim, Jun Sik Eom, Youyoung Choi, Yookyung Lee, Sung Sill Lee

**Affiliations:** 1Division of Applied Life Science (BK21), Gyeongsang National University, Jinju 52828, Korea; 2Institute of Agriculture & Life Science (IALS), Gyeongsang National University, Jinju 52828, Korea; 3Institute of Agriculture and Life Science & University-Centered Labs, Gyeongsang National University, Jinju 52828, Korea; 4Animal Nutrition and Physiology Team, National Institute of Animal Science, RDA, Jeonju 55365, Korea

**Keywords:** bamboo leave, waste recycling, in vitro batch culture, ruminal fermentation, methane, microbial population

## Abstract

**Simple Summary:**

Due to economic, environmental, and nutritional considerations, mitigating enteric methane production from ruminants is an important issue. Nutritionists have recently shown that feeding livestock natural feed additives could ameliorate this problem, due to the antimicrobial activities of the biologically active components in the additives. Bamboo is widely distributed in Asia and is currently being used in construction and paper pulp production, which results in a significant amount of bamboo leaves as by-products. The present study investigated whether bamboo leaves feeding can decrease methane production in ruminants. Here we found that bamboo leaves supplementation in vitro caused a 12.7–34.2% reduction in methane production after 12 and 48 h. Further studies are needed to demonstrate the effect of bamboo leaves supplementation in vivo, to determine its potential for mitigating methane production from ruminants.

**Abstract:**

Ruminants produce large amounts of methane as part of their normal digestive processes. Recently, feed additives were shown to inhibit the microorganisms that produce methane in the rumen, consequently reducing methane emissions. The objective of this study was to evaluate the dose–response effect of *Phyllostachys nigra* var. *henonis* (PHN) and *Sasa borealis* supplementation on in vitro rumen fermentation, methane, and carbon dioxide production, and the microbial population. An in vitro batch culture system was used, incubated without bamboo leaves (control) or with bamboo leaves (0.3, 0.6, and 0.9 g/L). After 48 h, total gas, methane, and carbon dioxide production decreased linearly with an increasing dose of bamboo leaves supplementation. The total volatile fatty acid, acetate, and acetate-to-propionate ratio were affected quadratically with increasing doses of bamboo leaves supplementation. In addition, propionate decreased linearly. Butyrate was increased linearly with increasing doses of PHN supplementation. The absolute values of total bacteria and methanogenic archaea decreased linearly and quadratically with an increasing dose of PHN treatment after 48 h. These results show that bamboo leaves supplementation can reduce methane production by directly affecting methanogenic archaea, depressing the metabolism of methanogenic microbes, or transforming the composition of the methanogenic community. These results need to be validated using in vivo feeding trials before implementation.

## 1. Introduction

Greenhouse gas emissions from the livestock industry are estimated to constitute 14.5–19% of global emissions [1]. Livestock enteric methane and carbon dioxide emissions contribute approximately 80% of the sector’s total emissions [2,3]. In addition, enteric methane emissions lose 6–12% of gross energy or 8–14% of digestible energy in ruminants’ energy intake [4,5]. Thus, reducing enteric methane emissions from livestock is an important aim for livestock nutritionists who consider the economy. Bambusoideae comprise 75 genera and 1642 bamboo species. It is one of the fastest-growing plants, with a growth rate of 7.5–100 cm/day and a short life cycle [6,7]. Bamboo is distributed across approximately 31.5 million hectares of land and is common in China, Korea, Japan, and Southeast Asia [8,9]. Bamboo is used for construction and paper pulp production, which results in significant amounts of bamboo leaves as a by-product. Therefore, the agro-industrial residues are useful as animal feed or additives to promote environmental sustainability of production systems [10,11]. The leaves of many bamboo species are used as feed for livestock in Japan, Pakistan, Nepal, and Africa. Previous studies have isolated biologically active compounds (flavonoids and polyphenols) from bamboo leaves, which have suitable properties for use in new applications [12,13]. These compounds have been found to possess useful characteristics, including antibacterial, antioxidant, and anti-inflammatory activities [14,15,16]. *Phyllostachys nigra* var. *henonis* (PHN) and *Sasa borealis* (SAB) have been shown to contain the biologically active compounds orientin, vitexin, luteolin, vittariflavone, and tricin [13] and isoorientin, tricin, and apigenin [12], respectively. Recently, many studies have shown that the biologically active compounds of by-products have the potential to mitigate methane production by modulating the ruminal microbial population [17,18]. In addition, recent studies have evaluated bamboo leaves as a feed additive for broiler chickens [19], a substitute for hay for sheep [20], a methane-reducing agent for goats [21], and a heat stress alleviator for Holstein dairy cows [22]. However, limited experimental data exist on the effects of bamboo leaves supplementation on livestock fermentation, methane production, and microbial population.

We hypothesized that bamboo leaves have biologically active compounds that would alter the microbial population of livestock to reduce enteric methane production without negatively affecting its fermentation. Therefore, we investigated the chemical composition of PHN and SAB and investigated their effects on in vitro methane production, rumen fermentation characteristics (e.g., pH, dry matter (DM) digestibility, and volatile fatty acid (VFA) production), and rumen microbial populations.

## 2. Materials and Methods

All experimental management and protocols related to animals used in this study were reviewed and approved by Gyeongsang National University Animal Research Ethical Committee (GNU-191011-E0050).

### 2.1. Collection of Bamboo Leaves Samples

Immature and mature leaves samples from eight trees of two bamboo (PHN and SAB) were harvested and pooled for leaves of each individual species over two consecutive months (May and June). After sampling, samples classified by species of origin were freeze-dried (Model PVTFD 10R, Ilshinbiobase CO., Ltd., Dongducheon, Korea) and stored in air-tight bags at −80 °C in a freezer until use for experiments.

### 2.2. Chemical Composition, Total Polyphenol, Flavonoid and Antioxidant Activity

The chemical composition of substrates used in in vitro experiments is shown in Table 1. Substrates and bamboo leaves were ground using a 1 mm screen with a Wiley mill for chemical, polyphenol, and flavonoid analysis and for in vitro experiments. The chemical composition of the substrate and bamboo leaves was analyzed by the AOAC [23], Van Soest [24], and Licitra [25] methods. Total polyphenol and flavonoid of bamboo leaves were analyzed by the Folin–Ciocalteu method [26] and the aluminum chloride method [27]. During the analysis, 2,2-diphenyl-1-picrylhydrazyl (DPPH) and 2,2′-azino-bis(3-ethylbenzothiazoline-6-sulfonic acid (ABTS) radical scavenging activity were used by modifying the method of Brand-Williams et al. [28] and Re et al. [29], respectively. The results were calculated as IC_50_, which is the minimum concentration required to scavenge 50% of the DPPH and ABTS radicals. The detailed procedures of such total polyphenol and flavonoid and radical scavenging activity analyses were the same as those of Lee et al. [18] and Seleshe et al. [30]. Substrates and bamboo leaves were analyzed in 3 replicates for chemical composition, total polyphenol, flavonoid, and antioxidant activity.

### 2.3. Animals and In Vitro Experiments

The whole rumen inoculum was obtained from two fistulated non-lactating Hanwoo (440 ± 20 kg)-fed Timothy hay (Table 1) and corn-based concentrate (moisture content, 13.01%; crude protein, 9.32%; ether extracts, 2.80%; crude ash, 1.35%; neutral detergent fiber (NDF), 35.14%, and acid detergent fiber (ADF), 5.04%) mixtures at a 6:4 ratio (DM basis), with free access to mineral block and water. The rumen inoculum was collected before morning feeding, transferred to the laboratory within 15 min, filtered with 4 layers of cheesecloth, and then mixed with McDougall’s buffer at a 1:2 ratio (vol/vol) to make a fluid mixture under constant O_2_-free N_2_ flushing at 39 °C. The bamboo leaves were used at four inclusion rates: 0 g/L (CON), 0.3 g/L (PHNL and SABL), 0.6 g/L (PHNM and SABM), and 0.9 g/L (PHNH and SABH). The fluid mixture (20 mL) was transferred to serum bottles (50 mL) under a stream of O_2_-free N_2_; substrates (0.3 g of Timothy) and bamboo leaves supplementation were placed into a nylon bag, which was sealed with butyl rubber and then crimped aluminum caps. Timothy was used as a control because it has a similar chemical composition to bamboo leaves and is a commonly used feed for ruminants. The amount of Timothy was chosen based on a previous study [18]. The serum bottles were incubated in a shaking incubator (120 rpm) at 39 °C for 6, 12, 24, 48, and 72 h. The in vitro batch culture experiments were replicated in four separate communities. Each in vitro batch culture experiment was a completely randomized block design and performed in two runs.

The actual trial design was:5 incubation times × 7 treatments × 4 repetitions × 2 runs

### 2.4. In Vitro Fermentation and Gas Profiles

After incubation, the headspace measurement pressure of the serum bottle was measured using a pressure transducer (Laurel Electronics, inc., Costa Mesa, CA, USA), and the total gas production was converted according to Theodorou et al. [31].

The equation used was:V = P/24.103 (n = 144, R^2^ = 0.999)
where V is the gas volume (mL), and P is the measured pressure (psi).

The headspace gas samples were collected in a vacuum tube, and then they were determined by gas chromatography (Agilent Technologies HP 5890, Waldbronn, Germany) equipped with a thermal conductivity detector on Carboxen 1006 PLOT capillary column (30 × 0.53 mm; Supelco, Bellefonte, PA, USA). The concentration of methane and carbon dioxide production was converted according to López et al. [32].

The equation used was:Methane and carbon dioxide production (mL)
= Concentration (mL/mL) × {Total gas (mL) + Headspace (30 mL)}

Ruminal pH of the fluid sample was determined by a pH meter (S210 SevenCompact, Mettler-Toledo, Greifensee, Switzerland) and collected and stored at −80 °C in a freezer for VFA and microbial population analysis. The VFA concentration was determined by high-performance liquid chromatography (HPLC; L-2200, Hitachi, Tokyo, Japan) following the method described by Adesogan et al. [33]. The apparent DM digestibility was determined following the nylon bag digestion method. Briefly, the residues were washed in a water bath and dried in a dry oven at 105 °C for 24 h.

### 2.5. The DNA Extraction and Real-Time Polymerase Chain Reaction

The genomic DNA of the 48 h rumen fluid sample (1.8 mL) was physically disrupted by the bead-beating method and extracted using a commercial DNA extraction kit (QIAamp Fast DNA stool Mini Kit, Qiagen, Valencia, CA, USA) according to the manufacturer’s recommendations. The genomic DNA yield, purity, and quality were determined by NanoDrop (Thermo Scientific, Wilmington, DE, USA).

Real-time polymerase chain reaction (RT-PCR) assays to determine the absolute abundance of total bacteria, methanogenic archaea, ciliate protozoa, fungi, hydrogen, and formate producing (*Ruminococcus albus* (*R. albus*), *Ruminococcus flavefaciens* (*R. flavefaciens*)), butyrate-producing (*Butyrivibrio fibrisolvens* (*B. fibrisolvens*), *Butyrivibrio proteoclasticus* (*B. proteoclasticus*)), succinate and propionate producing (*Fibrobacter succinogenes* (*F. succinogenes*), *Prevotella ruminicola* (*P. ruminicola*)) were performed using an RT-PCR Machine (CFX96 Real-Time system, Bio-Rad, Hercules, CA, USA). Information on sequence, size, efficiency, and PCR conditions references of target rumen microbes are shown in Table 2. Real-time PCR assays were measured in triplicate, a total reaction mixture of 20 μL. The total reaction mixture included 2 μL 10 × buffer (BioFACT, Daejeon, Korea), 0.5 μL 10 mM dNTP Mix (BioFACT, Daejeon, Korea), 1 μL forward primer (10 μM), 1 μL reverse primer (10 μM), 1 μL genomic DNA diluted 10-fold, 1 μL Evagreen (SolGent, Daejeon, Korea), 0.1 μL taq polymerase (BioFACT, Daejeon, Korea), and 13.4 μL PCR-grade water. For absolute quantification of each microbe, a standard curve obtains with a 10-fold dilution of plasmids containing the respective target gene sequence. For absolute quantification of each microorganism, a standard curve obtained by a plasmid containing each target gene sequence with 10-fold serial dilutions was used, and all the detailed manufacturing processes of each microbe plasmid were performed according to Kim et al. [34] and Hamid et al. [35]. All the reactions were performed in triplicate.

### 2.6. Statistical Analysis

Data were analyzed using the Interactive Matrix Language (IML) procedure and a completely randomized block design. Statistical analyses were conducted using SAS software (version 9.4, SAS Institute, Cary, NC, USA).

The statistical model used was:*Y_ijkl_* = *μ* + *α_i_* + *β_j_* + *γ_k_* + *ε_ijkl_*,
where *Y_ijkl_* represents the experimental data, µ the overall mean, *α_i_* the random effect of the fermentation run (I = 2), *β_j_* the fixed effect of dietary treatments, *γ_k_* the fixed effect of incubation times, and *ε_ijkl_* the unexplained random error. Polynomial contrasts were used to test the linear and quadratic effects of the treatments (i.e., bamboo leaves inclusion rate). Differences between means were identified using Tukey’s multiple comparison test, and significance was defined as *p* < 0.05.

## 3. Results

### 3.1. Chemical Composition and Antioxidant Activity

The chemical composition and antioxidant activities of the bamboo leaves of two cultivars (PHN and SAB) are shown in Table 3. The chemical compositions of the two cultivars are different: SAB had a higher percentage of crude protein, crude fiber, NDF, and NDICP, and a lower percentage of ether extract, crude ash, ADF, and ADFIP than PHN. In addition, SAB had higher total polyphenol and flavonoid content than PHN. The PHN and SAB cultivars contained 31.04 and 44.54 mg CE/g extract of polyphenols and 16.73 and 19.6 mg QE/g extract of flavonoids, respectively. Moreover, PHN leaves possessed ABTS, DPPH free radical scavenging properties, and higher antioxidant activities compared to SAB leave.

### 3.2. Effects of Bamboo Leaves on pH of Rumen Fluid and Dry Matter Digestibility

The pH of rumen fluid and DM digestibility results are shown in Table 4. The pH of rumen fluid ranged from 6.19 to 7.20 and increased linearly (*p* < 0.05) with increasing doses of bamboo leaves supplementation at 12 and 24 h. The highest pH was observed for SABH treatment at 12 h and PHNL treatment at 48 h (*p* < 0.05). Dry matter digestibility was affected quadratically (*p* < 0.05) with increasing doses of SAB supplementation at 12 and 24 h. Moreover, DM digestibility was significantly decreased following PHNL treatment (*p* < 0.05) compared to the CON group at 24 h.

### 3.3. Effects of Bamboo Leaves on Gas Profiles

In Table 5, total gas production decreased linearly (*p* < 0.01) with increasing doses of bamboo leaves supplementation, and the CON group was the highest (*p* < 0.05). The greatest decrease (*p* < 0.05) was observed in the PHN supplementation. After 12 and 48 h, methane and carbon dioxide production and proportions were decreased linearly (*p* < 0.05) with increasing doses of bamboo leaves supplementation.

### 3.4. Effects of Bamboo Leaves on Volatile Fatty Acids Profiles

The VFA profile results are shown in Table 6. After 24 h, the total VFA, acetate, and acetate-to-propionate ratio were affected linearly and quadratically (*p* < 0.05) with increasing doses of bamboo leaves supplementation. Moreover, propionate was decreased linearly (*p* < 0.05) with increasing doses of bamboo leaves supplementation. After 24 and 48 h, propionate and the acetate-to-propionate ratio were affected linearly (*p* < 0.05) with increasing doses of bamboo leaves supplementation, whereas butyrate was increased linearly (*p* < 0.05) with increasing doses of PHN supplementation.

### 3.5. Effects of Bamboo Leaves on the Abundance of Microbial Community

Regarding microbial counts, the abundance of total bacteria and methanogenic archaea were significantly lower (*p* < 0.05) following PHNM and PHNH treatments compared to the CON group, and were decreased linearly (*p* < 0.05) and quadratically (*p* < 0.05) with increasing doses of PHN supplementation at 48 h (Figure 1). Compared to the CON group, the abundance of ciliate protozoa was significantly higher (*p* < 0.05) in SABL treatment and was increased linearly (*p* < 0.05) and quadratically (*p* < 0.05) with increasing doses of SAB supplementation. No significant changes in ciliate protozoa abundance were observed in PHN supplementation (*p* < 0.05) change. None of the bamboo leaves supplementations significantly affected (*p* < 0.05) the abundance of fungi. The abundance of *R. albus*, *R. flavefaciens*, *B. proteoclasticus*, and *B. fibrisolvens* was decreased linearly (*p* < 0.05) with increasing doses of PHN supplementation, while SAB supplementation had no effect (*p* > 0.05) (Figure 2). The abundance of *P. ruminicola* was decreased linearly (*p* < 0.05) with increasing doses of PHN supplementation, while the abundance of *F. succinogenes* was not affected (*p* > 0.05).

## 4. Discussion

The leaves of various trees contain biologically active compounds [42]. Of the plant’s secondary metabolites, polyphenols and flavonoids are abundant in bamboo leaves. Plant secondary metabolites may improve ruminal fermentation, stimulate microbial metabolism, modulate the microbial population, and mitigate methane production in the rumen [43,44,45,46]. Therefore, nutritious bamboo leaves containing biologically active compounds can potentially be used as feed additives for ruminants. Teh et al. [47] reported a positive correlation between flavonoid content and antioxidant activity (i.e., DPPH and ABTS radical scavenging activity). However, in our study, SAB leaves had higher flavonoid levels and lower antioxidant activities compared to PHN leaves.

Based on the in vitro batch cultures, the pH of rumen fluid increased linearly in a dose-dependent manner. This result was supported by the observation that polyphenols and flavonoids slowed the digestion of fiber in the rumen [48]. Most acetate- and butyrate-producing fibrolytic bacteria are sensitive to the pH of rumen fluid below 5.0–5.5 [49]. As confirmed in this study, bamboo leaves supplementation negatively affected fibrolytic bacteria (*R. albus*, *R. flavefaciens*, and *F. succinogenes*). Furthermore, pH changes are associated with rumen digestion and fermentation, because they can occur through the accumulation of VFA. Here, bamboo leaves supplementation induced a shift in the VFA pattern and affected the pH. Notably, bamboo leaves supplementation caused a shift in ruminal fermentation to a greater acetate concentration and acetate-to-propionate ratio and a reduced concentration of propionate after 24 h. This would have a negative effect on methane mitigation, because the formation of acetate results in the production of hydrogen [50], while the formation of propionate creates a hydrogen sink, reducing the availability of hydrogen to methanogen [51]. In addition, decreased ruminal propionate production should decrease blood glucose concentration via gluconeogenesis and then decrease milk lactose [52]. The DM digestibility by bamboo leaves supplementation was approximately 4.9–11.2% improved than the CON group, which may be influenced by plant secondary metabolites [53].

Methane production was not affected by bamboo leaves supplementation after 24 h incubation, but 12 and 48 h reduced approximately 12.7–34.2% more than the CON group, without compromising total VFA production. Jafari et al. [21] reported that different amounts of bamboo leaves (15, 25, and 50% of Alfalfa hay, which was used as substrate in Jafari’s study and was replaced by bamboo leaves) reduced methane production by 29–62%. The greater reduction observed here may be due to differences in substrate addition and replacement, or the study by Jafari et al. [21] showed a linearly decreasing trend in total VFA with increasing doses of bamboo leaves, or both. Furthermore, it may be because the chemical compositions of plants are influenced by their geographic origin, harvest stage, and age [54,55]. The lower abundance of methanogenic archaea following PHN supplementation may support this result. This result is in agreement with a previous study that reported a decreased number of methanogenic archaea in response to direct inhibition of polyphenols and flavonoids of PHN [56]. SAB supplementation did not affect the number of methanogenic archaea but did slightly decrease methane production, which may be due to the depressed metabolism of methanogenic microbes, the transformed composition of the methanogenic community, or both [57]. In addition, the lower abundance of hydrogen-producing bacteria (*R. albus* and *R. flavefaciens*) was reduced by PHN supplementation. This result is supported by flavonoids having antibacterial properties against gram-positive bacteria [58]. Wann et al. [59] observed that fibrolytic bacteria numbers increased linearly in vivo with an increase in the concentration of bamboo leaves (from 0–150 g/d per head). Our results did not increase the abundance of fibrolytic organisms (Fungi, *Ruminococcus*, and *Fibrobacter*), which may be due to the drying method of the bamboo leaves used in this study. Freeze drying has a higher antioxidant capacity for secondary metabolites than sun drying [60]. This result is also supported by the lower abundance of hydrogen-producing bacteria (*R. albus* and *R. flavefaciens*) following PHN supplementation.

Previous studies [61,62] have observed a shift to butyrate production instead of propionate production following natural feed additives supplementation, which is in keeping with our findings following PHN supplementation at 48 h. However, the abundance of butyrate-producing bacteria (*B. proteoclasticus* and *B. fibrisolvens*) was not decreased, suggesting that this result is due to the changed butyrate-producing bacteria that produce another metabolite, as reported by Oh et al. [63], or increased butyrate production at the expense of acetate, or both. According to Janssen [64], since the butyrate production pathway competes for dihydrogen, more butyrate production reduces hydrogen, which is accompanied by a reduction in methane production. In addition, rumen epithelial cells use butyrate as a major energy source, and butyrate stimulates its development to improve feed utilization [65]. The reason for the mitigated methane production following SAB supplementation is not clear. However, this result may be explained by the reduction of carbon dioxide, an important substrate of methanogenic bacteria, during methane production following SAB supplementation.

## 5. Conclusions

Bamboo leaves are rich in polyphenols and flavonoids and possess antioxidant activities. Experimental data on the effects of bamboo leaves supplementation on rumen fermentation, methane production, and microbial populations are limited. Based on the data obtained from the present study, it could be concluded that bamboo leaves did not negatively affect DM digestibility and VFA production but did reduce total gas, methane, and carbon dioxide production and alter the microbial population. Dietary inclusion of bamboo leaves could serve as a promising additive for mitigating enteric methane production, however; further studies are needed to demonstrate the effect of bamboo leaves supplementation in vivo for mitigating enteric greenhouse gas production from ruminants.

## Figures and Tables

**Figure 1 animals-12-02222-f001:**
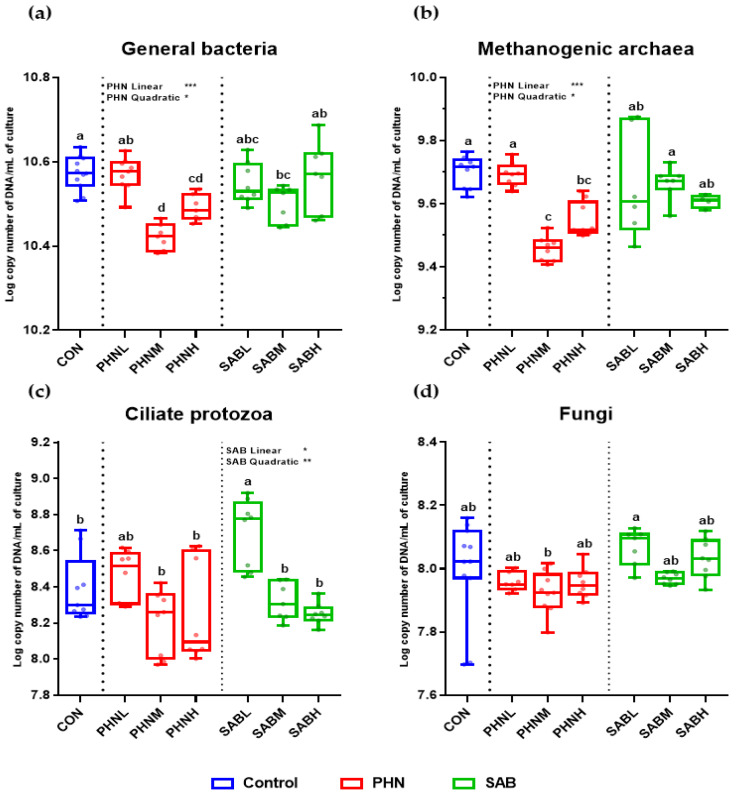
Effects of bamboo leaves addition on the population of general bacteria (**a**), methanogenic archaea (**b**), ciliate protozoa (**c**), and fungi (**d**) (Log copy number of DNA/mL of rumen fluid) at 48 h incubation time. Error bars are standard error of the mean (*n* = 4). ^abcd^ Means with different superscripts in the same row indicate significant differences (*p* < 0.05). Orthogonal contrasts for linear and quadratic effects were considered statistically significant at * *p* < 0.050, ** *p* < 0.010, and *** *p* < 0.001.

**Figure 2 animals-12-02222-f002:**
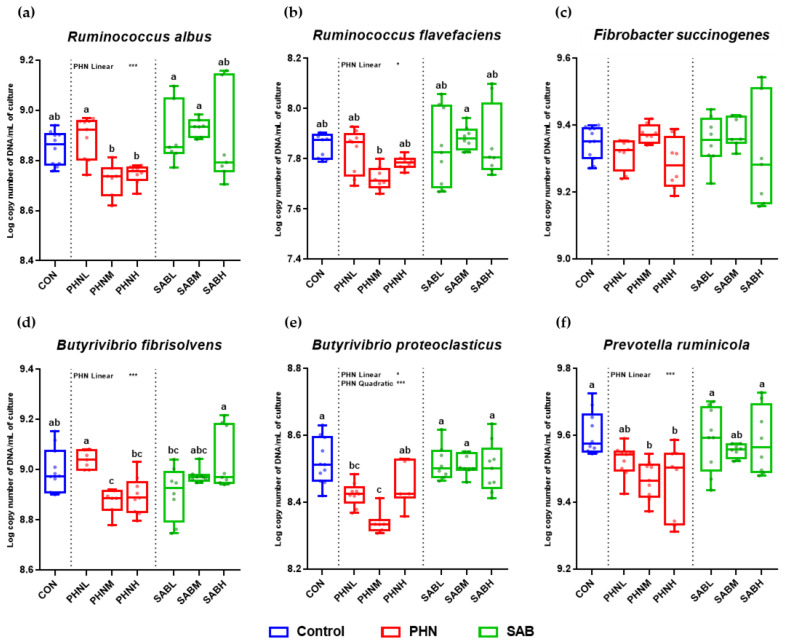
Effects of bamboo leaves addition on the population of *Ruminococcus albus* (**a**), *flavefaciens* (**b**), *Fibrobacter succinogenes* (**c**), *Butyrivibrio fibrisolvens* (**d**), *proteoclasticus* (**e**), and *Prevotella ruminicola* (**f**) (Log copy number of DNA/mL of culture) at 48 h incubation time. Error bars are standard error of the mean (*n* = 4). ^abc^ Means with different superscripts in the same row indicate significant differences (*p* < 0.05). Orthogonal contrasts for linear and quadratic effects were considered statistically significant at * *p* < 0.050 and *** *p* < 0.001.

**Table 1 animals-12-02222-t001:** Chemical composition of control diet (dry matter basis, %).

Items ^1^	Timothy Hay	SD ^2^
Chemical composition		
Dry matter	94.11	0.64
Ether extract	5.48	0.12
Crude protein	10.46	0.68
Crude ash	5.93	0.06
Crude fiber	30.11	0.23
NDF	62.27	0.22
ADF	38.14	0.75

^1^ NDF, neutral detergent fiber; ADF, acid detergent fiber. ^2^ SD, standard deviation of 3 replicates.

**Table 2 animals-12-02222-t002:** Primers for real-time polymerase chain reaction assays.

Target Species	Primer ^1^	Sequence	Size (bp) ^2^	Efficiency ^3^	References
General bacteria	F	CGGCAACGAGCGCAACCC	130	1.98	[36,37]
	R	CCATTGTAGCACGTGTGTAGCC			
Methanogenic archaea	F	GAGGAAGGAGTGGACGACGGTA	232	1.92	[38]
	R	ACGGGCGGTGTGTGCAAG			
Ciliate protozoa	F	GCTTTCGWTGGTAGTGTATT	223	1.91	[38]
	R	CTTGCCCTCYAATCGTWCT			
Fungi	F	GAGGAAGTAAAAGTCGTAACAAGGTTTC	120	1.83	[36]
	R	CAAATTCACAAAGGGTAGGATGATT			
*Ruminococcus albus*	F	CCCTAAAAGCAGTCTTAGTTCG	176	1.93	[39]
	R	CCTCCTTGCGGTTAGAACA			
*Ruminococcus flavefaciens*	F	CGAACGGAGATAATTTGAGTTTACTTAGG	132	1.81	[36]
	R	CGGTCTCTGTATGTTATGAGGTATTACC			
*Fibrobacter succinogenes*	F	GTTCGGAATTACTGGGCGTAAA	121	1.88	[36]
	R	CGCCTGCCCCTGAACTATC			
*Butyrivibrio fibrisolvens*	F	ACCGCATAAGCGCACGGA	65	1.89	[40]
	R	CGGGTCCATCTTGTACCGATAAAT			
*Butyrivibrio proteoclasticus*	F	TCCGGTGGTATGAGATGGGC	185	2.22	[41]
	R	GTCGCTGCATCAGAGTTTCCT			
*Prevotella ruminicola*	F	GCGAAAGTCGGATTAATGCTCTATG	78	2.04	[37]
	R	CCCATCCTATAGCGGTAAACCTTTG			

^1^ F, forward; R, reverse. ^2^ bp, base pair. ^3^ efficiency is calculated as [10^−1^/slope].

**Table 3 animals-12-02222-t003:** Chemical composition and antioxidant activity of bamboo leaves (DM basis, %).

Items ^1^	*Phyllostachys nigra* Var. *Henonis*	SD ^2^	*Sasa borealis*	SD
Chemical composition				
Dry matter (DM)	45.39	0.24	54.84	0.22
Ether extract	2.49	0.56	0.65	0.15
Crude protein (CP)	13.15	0.32	14.82	0.20
Crude ash	12.12	0.11	7.99	0.05
Crude fiber	21.08	0.07	24.19	0.38
NDF	65.6	0.84	70.58	0.88
ADF	38.02	0.30	35.57	0.27
NDICP (CP basis, %)	12.83	0.06	14.26	0.07
ADICP (CP basis, %)	5.26	0.13	2.92	0.09
Total polyphenol (mg CE/g extract)	31.04	5.83	44.54	6.07
Total flavonoid (mg QE/g extract)	16.73	0.52	19.6	0.49
IC_50_ for DPPH (µg/mL)	115.58	6.06	125.43	3.83
IC_50_ for ABTS (µg/mL)	47.43	0.61	54.17	2.53

^1^ NDF, neutral detergent fiber; ADF, acid detergent fiber; NDICP, neutral detergent insoluble crude protein; ADICP, acid detergent insoluble crude protein; CE, catechin equivalent; QE, quercetin equivalent; IC_50_, half maximal inhibitory concentration. ^2^ SD, standard deviation of 3 replicates.

**Table 4 animals-12-02222-t004:** Effects of bamboo leaves on pH and dry matter digestibility in in vitro incubation.

Incubation Time (h)	CON ^1^	Treatment ^2^						SEM ^3^	*p* Value	Contrasts ^4^			
		PHN			SAB					PHN Set		SAB Set	
		PHNL	PHNM	PHNH	SABL	SABM	SABH			L	Q	L	Q
pH													
12	6.91 ^d^	7.03 ^c^	7.09 ^bc^	7.08 ^bc^	7.14 ^ab^	7.12 ^abc^	7.20 ^a^	0.02	<0.0001	***	*	***	***
24	6.59 ^b^	6.72 ^a^	6.69 ^a^	6.71 ^a^	6.77 ^a^	6.72 ^a^	6.73 ^a^	0.02	0.0003	***	**	**	**
48	6.19 ^b^	6.30 ^a^	6.27 ^ab^	6.25 ^ab^	6.23 ^ab^	6.20 ^ab^	6.20 ^ab^	0.02	0.0218	ns	*	ns	ns
Dry matter digestibility (%)													
6	31.1	31.4	31.5	32.4	34.3	33.4	33.3	1.32	0.5560	ns	ns	ns	ns
12	38.7 ^b^	43.0 ^ab^	40.8 ^ab^	42.6 ^ab^	42.9 ^ab^	43.5 ^a^	42.0 ^ab^	1.01	0.0392	ns	ns	*	*
24	53.1	56.2	53.9	57.0	56.9	56.6	54.1	1.32	0.2069	ns	ns	ns	*
48	65.5 ^a^	61.8 ^b^	64.9 ^ab^	63.5 ^ab^	64.0 ^ab^	64.6 ^ab^	65.4 ^ab^	0.79	0.0466	ns	ns	ns	ns
72	68.7	66.4	67.5	66.4	67.3	67.8	69.3	0.85	0.1797	ns	ns	ns	ns

^1^ CON, control (without bamboo leaves). ^2^ PHN, *Phyllostachys nigra* var. *henonis*; SAB, *Sasa borealis*; PHN and SAB dose levels of 0.3 g/L (PHNL and SABL), 0.6 g/L (PHNM and SABM), and 0.9 g/L (PHNH and SABH). ^3^ SEM, standard error of the mean. ^4^ orthogonal contrasts for L, linear and Q, quadratic effects. The levels of significance were assigned as follow, ns, non-significant; *, *p* < 0.050; **, *p* < 0.010; ***, *p* < 0.001. ^abc^ Means (*n* = 4) with different superscripts in the same column differ significantly (*p* < 0.05).

**Table 5 animals-12-02222-t005:** Effects of bamboo leaves on gas profiles in in vitro incubation.

Incubation Time (h)	CON ^1^	Treatment ^2^						SEM ^3^	*p* Value	Contrasts ^4^			
		PHN			SAB					PHN Set		SAB Set	
		PHNL	PHNM	PHNH	SABL	SABM	SABL			L	Q	L	Q
Total gas (mL∙g^−1^ DM ^5^)													
12	87.0 ^a^	82.1 ^b^	82.0 ^b^	81.2 ^b^	82.2 ^b^	81.2 ^b^	77.5 ^c^	0.77	<0.0001	***	*	***	ns
24	121 ^a^	115 ^ab^	115 ^abc^	112 ^bc^	116 ^ab^	115 ^bc^	109 ^c^	1.31	0.0002	***	ns	***	ns
48	156 ^a^	147 ^ab^	147 ^b^	146 ^b^	151 ^ab^	147 ^b^	147 ^ab^	1.97	0.0146	**	*	**	ns
Methane (mL∙g^−1^ DM)													
12	7.40 ^a^	5.84 ^ab^	4.87 ^b^	5.83 ^ab^	6.26 ^ab^	6.15 ^ab^	5.57 ^b^	0.37	0.0051	**	*	***	ns
24	14.8	12.7	13.7	14.6	14.2	14.1	13.8	0.79	0.6162	ns	*	ns	ns
48	23.7 ^a^	20.7 ^ab^	19.1 ^b^	18.5 ^b^	19.1 ^ab^	19.7 ^b^	19.4 ^ab^	0.73	0.0010	***	ns	**	*
Methane/Total gas (%)													
12	8.52 ^a^	7.12 ^ab^	5.94 ^b^	7.19 ^ab^	7.62 ^ab^	7.57 ^ab^	7.19 ^ab^	0.49	0.0488	ns	*	*	ns
24	12.2	11.0	11.9	13.0	12.3	12.3	12.7	0.64	0.4616	ns	*	ns	ns
48	15.2 ^a^	14.1 ^ab^	13.0 ^ab^	12.7 ^b^	12.6 ^b^	13.5 ^ab^	13.9 ^ab^	0.53	0.0249	**	ns	ns	*
Carbon dioxide (mL∙g^−1^ DM)													
12	35.5 ^a^	27.3 ^ab^	22.6 ^b^	26.7 ^b^	30.7 ^ab^	28.5 ^ab^	27.2 ^ab^	1.89	0.0044	**	**	**	ns
24	55.0	45.9	49.0	52.1	51.8	50.9	49.4	2.90	0.4603	ns	*	ns	ns
48	80.0 ^a^	70.8 ^ab^	63.5 ^b^	60.7 ^b^	63.5 ^b^	65.5 ^b^	67.6 ^b^	2.52	0.0005	***	ns	**	*
Carbon dioxide/Total gas (%)													
12	40.8 ^a^	33.3 ^ab^	27.6 ^b^	32.9 ^ab^	37.4 ^ab^	35.1 ^ab^	35.0 ^ab^	2.39	0.0272	*	*	ns	ns
24	45.7	39.8	42.7	46.6	44.6	44.3	45.4	2.44	0.5343	ns	*	ns	ns
48	51.3 ^a^	48.1 ^ab^	43.3 ^ab^	41.6 ^b^	42.0 ^b^	44.7 ^ab^	46.1 ^ab^	1.85	0.0123	***	ns	ns	*

^1^ CON, control (without bamboo leaves). ^2^ PHN, *Phyllostachys nigra* var. *henonis*; SAB, *Sasa borealis*; PHN and SAB dose levels of 0.3 g/L (PHNL and SABL), 0.6 g/L (PHNM and SABM), and 0.9 g/L (PHNH and SABH). ^3^ SEM, standard error of the mean. ^4^ orthogonal contrasts for L, linear and Q, quadratic effects. ^5^ DM, dry matter. The levels of significance were assigned as follow, ns, non-significant; *, *p* < 0.050; **, *p* < 0.010; ***, *p* < 0.001. ^abc^ Means (*n* = 4) with different superscripts in the same column differ significantly (*p* < 0.05).

**Table 6 animals-12-02222-t006:** Effects of bamboo leaves on volatile fatty acid profiles in in vitro incubation.

Incubation Time (h)	CON ^1^	Treatment ^2^						SEM ^3^	*p* Value	Contrasts ^4^			
		PHN			SAB					PHN Set		SAB Set	
		PHNL	PHNM	PHNH	SABL	SABM	SABH			L	Q	L	Q
Total VFA (mM)													
12	54.8	54.3	54.7	54.8	54.9	54.6	54.5	0.479	0.9634	ns	ns	ns	ns
24	73.4 ^bc^	73.7 ^bc^	76.6 ^ab^	73.1 ^c^	78.0 ^a^	77.4 ^a^	75.0 ^abc^	0.741	0.0002	ns	**	ns	***
48	78.4	78.2	78.4	78.2	80.3	80.5	83.6	1.259	0.0534	ns	ns	*	ns
Acetate (mM)													
12	35.3	35.0	35.5	35.5	35.2	35.0	34.9	0.354	0.7824	ns	ns	ns	ns
24	45.6 ^c^	47.4 ^bc^	49.7 ^ab^	46.8 ^bc^	52.1 ^a^	50.9 ^a^	48.9 ^abc^	0.765	<0.0001	*	**	*	***
48	51.5 ^b^	51.2 ^b^	51.3 ^b^	51.7 ^ab^	54.3 ^ab^	53.0 ^ab^	56.4 ^a^	1.017	0.0115	ns	ns	*	ns
Propionate (mM)													
12	13.0 ^ab^	12.7 ^ab^	12.5 ^b^	12.8 ^ab^	13.1 ^a^	12.8 ^ab^	12.5 ^ab^	0.127	0.0188	ns	*	*	ns
24	19.3 ^a^	18.4 ^bc^	18.8 ^b^	18.3 ^c^	18.0 ^c^	18.4 ^bc^	18.1 ^c^	0.108	<0.0001	***	ns	***	***
48	18.8 ^ab^	17.1 ^c^	17.0 ^c^	17.2 ^bc^	17.8 ^abc^	18.0 ^abc^	19.0 ^a^	0.370	0.0023	*	*	ns	*
Butyrate (mM)													
12	6.43	6.64	6.74	6.57	6.65	6.80	7.07	0.140	0.1014	ns	ns	**	ns
24	8.43 ^a^	7.88 ^b^	8.09 ^ab^	7.94 ^ab^	7.87 ^b^	8.07 ^ab^	8.05 ^ab^	0.117	0.0463	*	ns	ns	*
48	8.09 ^b^	9.94 ^a^	10.12 ^a^	9.29 ^ab^	8.17 ^b^	9.40 ^ab^	8.20 ^b^	0.312	0.0002	**	***	ns	ns
Acetate-to-propionate ratio													
12	2.71 ^b^	2.77 ^ab^	2.84 ^a^	2.78 ^ab^	2.69 ^b^	2.72 ^b^	2.78 ^ab^	0.026	0.0064	*	ns	*	ns
24	2.36 ^c^	2.57 ^bc^	2.64 ^b^	2.56 ^bc^	2.90 ^a^	2.77 ^ab^	2.71 ^ab^	0.046	<0.0001	**	**	**	***
48	2.74 ^b^	2.99 ^ab^	3.02 ^ab^	3.02 ^ab^	3.06 ^a^	2.94 ^ab^	2.96 ^ab^	0.064	0.0445	**	*	ns	ns

^1^ CON, control (without bamboo leaves). ^2^ PHN, *Phyllostachys nigra* var. *henonis*; SAB, *Sasa borealis*; PHN and SAB dose levels of 0.3 g/L (PHNL and SABL), 0.6 g/L (PHNM and SABM), and 0.9 g/L (PHNH and SABH). ^3^ SEM, standard error of the mean. ^4^ orthogonal contrasts for L, linear and Q, quadratic effects. The levels of significance were assigned as follow, ns, non-significant; *, *p* < 0.050; **, *p* < 0.010; ***, *p* < 0.001. ^abc^ Means (*n* = 4) with different superscripts in the same column differ significantly (*p* < 0.05).

## Data Availability

Not applicable.

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
