# Peer review of "Dose–Response Effects of Bamboo Leaves on Rumen Methane Production, Fermentation Characteristics, and Microbial Abundance In Vitro"

_animals, 2022, doi:10.3390/ani12172222_

Round 1

Reviewer 1 Report

In general a well written manuscript on a very important topic, methods and results are scientifically sound. In the Discussion there are parts missing: interpretation of gas production, shifts in VFA composition in regards to in vivo situation and milk production (big concern for farmers) is lacking. Also comment on inclusion rate (see comment below) translated into in vivo situation is missing. Dry matter disappearance or digestibility of NDF and OM would have been interesting parameters to include, if not available then the overall gas production and implications of these results should be described in more detail in discussion part.

Line 4: “In in vitro” in the title seems to be a double mentioning, just in vitro would be sufficient

Line 23: Mentioning of in vitro is missing

Line 83: word is missing, maybe stored

Line 87: are shown

Line 90: please mention methods

Line 99: is there a specific reason to state the standard error instead of the standard deviation?

Line 118: actual trial design is not clearly described, a scheme/graph or table would help understand setup

Line 132: Delete )

Line 191: as it was an in vitro experiment I think the noun ruminal pH would be misleading, clearer would be: the pH of rumen fluid

Line 198: 0h measurements of rumen fluid are not included, to my understanding also not as a co-variate, correct? If so then the 0h measurements should be included

Line 273: Possible implication of lower propionate production on milk production is not mentioned here and should be described

General: Translated to in vivo situation and assuming a rumen volume of 100l inclusion rates would range from 30g to 90g Bamboo leaves per head and day à feasibility of inclusion rate in vivo situation given?

Author Response

Response to Reviewer 1 Comments

In general a well written manuscript on a very important topic, methods and results are scientifically sound. In the Discussion there are parts missing: interpretation of gas production, shifts in VFA composition in regards to in vivo situation and milk production (big concern for farmers) is lacking. Also comment on inclusion rate (see comment below) translated into in vivo situation is missing. Dry matter disappearance or digestibility of NDF and OM would have been interesting parameters to include, if not available then the overall gas production and implications of these results should be described in more detail in discussion part.

Point 1: Line 4: “In in vitro” in the title seems to be a double mentioning, just in vitro would be sufficient

Response 1: Thank you for reviewer’s comment! We have changed “in in vitro” to “in vitro”. (Line 4)

Point 2: Line 23: Mentioning of in vitro is missing

Response 2: Thank you very much! We have added “in vitro”. (Line 24)

Point 3: Line 83: word is missing, maybe stored
Response 3: Thank you very much! We have added “stored”. (Line 91)

Point 4: Line 87: are shown

Response 4: Thanks a lot! We have changed “are show” to “is shown”. (Line 95)

Point 5: Line 90: please mention methods

Response 5: Thank you for reviewer’s comment! We have added mention method. (Line 98)

Point 6: Line 99: is there a specific reason to state the standard error instead of the standard deviation?

Response 6: Thank you for reviewer’s comment! In general, standard errors are used to evaluate significance using statistical analysis. We calculated the standard deviation from Table 1 and 3, but marked there as standard error. Therefore, we have changed “Standard error” to “Standard deviation”. (Table 1 and 3) We hope this response will be acceptable to reviewer.

Point 7: Line 118: actual trial design is not clearly described, a scheme/graph or table would help understand setup

Response 7: Thank you for reviewer’s comment! We have added the actual trial design from table. (Line 131)

Point 8: Line 132: Delete )

Response 8: Thank you for reviewer’s comment! We have changed “)” to “A”. (Line 146).

Point 9: Line 191: as it was an in vitro experiment I think the noun ruminal pH would be misleading, clearer would be: the pH of rumen fluid

Response 9: Thank you for reviewer’s comment! We have changed “ruminal pH” to “pH of rumen fluid”. (Line 205, 206, 207 and 279)

Point 10: Line 198: 0h measurements of rumen fluid are not included, to my understanding also not as a co-variate, correct? If so then the 0h measurements should be included

Response 10: We have agreed with the reviewer’s comment! Unfortunately, however we did not analyze the pH of rumen fluid. In the next experiment, we will analyze the 0h measurements of rumen fluid. We hope this response will be acceptable to reviewer.

Point 11: Line 273: Possible implication of lower propionate production on milk production is not mentioned here and should be described

Response 11: Thank you for reviewer’s comment! We have added the effect of lower propionate production on milk production. (Line 292-294)

“In addition, decreased ruminal propionate production should decrease blood glucose concentration via gluconeogenesis and then decrease milk lactose.”

Reference : Stein, D.R.; Allen, D.T.; Perry, E.B.; Bruner, J.C.; Gates, K.W.; Rehberger, T.G.; Mertz, K.; Jones, D.; Spicer, L.J. Effects of Feeding Propionibacteria to Dairy Cows on Milk Yield, Milk Components, and Reproduction. J. Dairy Sci. 2006, 89, 111–125.

Point 12: General: Translated to in vivo situation and assuming a rumen volume of 100l inclusion rates would range from 30g to 90g Bamboo leaves per head and day à feasibility of inclusion rate in vivo situation given?

Response 12: In my opinion, it would be possible to 30g to 90g bamboo leaves per head and day. However, there are many variables in the in vivo experiment, so the results are no certainty that will be the same as in vitro results.

Thank you for investing your time and reviewing the manuscript.

Reviewer 2 Report

The subject matter is certainly interesting and topical. The aim is to test different doses of bamboo leaves in order to assess their effect on methane production in vitro. However, there are several relevant corrections to be made throughout the text.  The introduction is not consistent. The experimental scheme is unclear, in particular why use a forage as a control substrate. The discussion definitely needs to be improved.

Author Response

Response to Reviewer 2 Comments

The subject matter is certainly interesting and topical. The aim is to test different doses of bamboo leaves in order to assess their effect on methane production in vitro. However, there are several relevant corrections to be made throughout the text.

Point 1: The introduction is not consistent.

Point 2: The experimental scheme is unclear, in particular why use a forage as a control substrate.

Point 3: The discussion definitely needs to be improved.

Thank you for reviewer’s comment!

Response 1: We have added sentences to make the introduction consistent. (Line 58-61)

“Bamboo is used for construction and pulp production, which results in significant amounts of bamboo leaves as a by-product. Therefore, the agro-industrial residues are useful as animal feed or additives to promote environmental sustainability of production systems.”

Reference : Véras, R.M.L.; Gois, G.C.; Nascimento, D.B.; Magalhães, A.L.R.; Teodoro, A.L.; Pinto, C.S.; Oliveira, L.P.; Campos, F.S.; Andrade, A.P.; Lima, I.E. Potential Alternative Feed Sources for Ruminant Feeding from the Biodiesel Production Chain By-Products. S. Afr. J. Anim. Sci. 2020, 50, 69–77.

Serrapica, F.; Masucci, F.; Raffrenato, E.; Sannino, M.; Vastolo, A.; Barone, C.M.A.; Di Francia, A. High Fiber Cakes from Mediterranean Multipurpose Oilseeds as Protein Sources for Ruminants. Animals 2019, 9, 918.

Response 2: We have added the actual trial design from table (Line 131) and why timothy was used as a control substrate (Line 125-127).

  1. The actual trial design was:

5 incubation times × 7 treatments × 4 repetitions × 2 runs

  1. “Timothy was used as a control because it has similar chemical composition to bamboo leaves and is a commonly used feed for ruminants.”

Response 3: We added some sentences to improve the discussion (Line 282-284 and 292-294).

“Most acetate and butyrate producing fibrolytic bacteria are sensitive to the pH of rumen fluid below 5.0—5.5 [48]. As confirmed in this study, bamboo leaves supplementation negatively affected fibrolytic bacteria (R. albus, R. flavefaciens, and F. succinogenes).”

Reference : Hoover, W.H. Chemical Factors Involved in Ruminal Fiber Digestion. J. Dairy Sci. 1986, 69, 2755–2766.

“In addition, decreased ruminal propionate production should decrease blood glucose concentration via gluconeogenesis and then decrease milk lactose.”

Reference : Stein, D.R.; Allen, D.T.; Perry, E.B.; Bruner, J.C.; Gates, K.W.; Rehberger, T.G.; Mertz, K.; Jones, D.; Spicer, L.J. Effects of Feeding Propionibacteria to Dairy Cows on Milk Yield, Milk Components, and Reproduction. J. Dairy Sci. 2006, 89, 111–125.

We hope this response will be acceptable to reviewer. Thank you for investing your time and reviewing the manuscript.

Reviewer 3 Report

animals-1853702

GENERAL REMARKS

Dear authors,

I have evaluated your manuscript identified as animals-1853702. The manuscript addresses the potential of bamboo foliage as an alternative to hay in the feeding of ruminants. Considering that the literature on this subject is scarce, your contribution could be greatly appreciated. That said, I found some inaccuracies as well as I identified some room for improvement, for some parts. My main criticisms concern the materials and methods, specifically paragraphs 2.1 and 2.2, in which the description of the sampling and analysis methods should be better detailed and described, in my opinion. Even the introduction has some critical issues and I think it can also be revised to make the work more captivating and more visible for readers. On the other hand, except for a few minor details, both the results and the discussions are well organized and detailed in my opinion. The conclusions are fine. My specific comments are listed below, point by point. I hope my contribution will improve the quality of the manuscript. Good work.

SPECIFIC COMMENTS

L 17-18: the purposes that the authors highlighted (i.e., to mitigate enteric emissions) are true for all ruminants, rather than just cattle. Since the manuscript is aimed at an exploratory in vitro study, it is my opinion that it is more useful to generalize by referring to all ruminants. Thanks.

L 23: please, replace “livestock” with “ruminants”. Thanks.

L 24: please, replace “suppression” with “reduction”. Thanks.

L 25: I suggest recasting the sentence avoiding the syntagma “in in vivo”. Thanks.

L 29-30: in my opinion, the two bamboo species tested in the study should be already mentioned in this part of the text. Thanks.

L 43: in my opinion, "waste recycling" or "circular economy" can be included in the keywords. Thanks

L 47-48: authors are requested to check punctuation. In addition, the phrase "Greenhouse gas emissions from the livestock industry are estimated to constitute 14.5 — 19% of global emissions" should be supported by adequate references. Thanks.

L 52: please, delete the comma between “economy” “and the environment”. Moreover, the article on the environment is superfluous Thanks.

L 53-57: recovery and valorization of agro-industrial residues are currently indicated as key factors for the development of the circular economy models and to promote the environmental sustainability of production systems. International scientific opinion is very interested in these issues, which the authors have just mentioned in the simple summary (i.e., 21-22). In my opinion, however, it should be stressed more throughout the text and, particularly, in the introduction. I suppose that this view is valid because the cultivation of bamboo is mainly aimed at the production of lignocellulosic biomass, in which the leaves constitute a waste of the supply chain. So, I would suggest to the authors, albeit briefly, mention the referenced supply chains and the foliage as their by-products and the possibility of reuse of residues between agricultural supply chains. In this regard, I suggest using manuscript doi:10.3390/ani9110918 as a template (especially the introduction), which I strongly recommend using as a reference. Thanks

L 55: the area covered with bamboo that the authors report refers only to China or to another geographical scale. The authors are requested to clarify, thanks.

L 57: the use of the term "previous studies" implies that reference is made to the studies referred to. However, no indication is given in this regard. Authors are requested to rearrange the sentence by noting the studies to which they refer. Thanks.

L 69-71: on what basis did the authors formulate their hypothesis? From the literature data reported above the hypothesis cannot be guessed. I think the authors should better formulate the sentence, since what is reported as a hypothesis seems to coincide more with an expected result. Alternatively, greater clarification of the state of the art could help to understand the hypothesis formulated by the authors. Thanks.

L 71-74: perhaps it would be better if the authors avoided generalizing. In fact, more than just bamboo leaves, the authors have analyzed leaves of two specific species, which in my opinion should be clearly mentioned. Thanks.

L 80-84: in my opinion, there are several inconsistencies in this sentence. First, it is unclear whether the leaves of eight plants were used for each species tested or whether the eight plants refer to both species (4 plants/species). It is also not clear whether the "young and mature" leaves (the term "young" is not very technical and should be replaced with immature ") were harvested progressively over the indicated time (May and June) and if so, in what sequence. Given the differences in chemical composition between the leaves of the two species, these aspects are not marginal. I'm not sure if the leaves collected were grouped by type of leaf within each species or simply by species of origin. The statement "were harvested and pooled for each species of leaves" creates perplexity in this regard. Considering what is reported below [materials and methods (paragraphs 2.3 and 2.4) and results] I have not noticed any analysis differentiated by the type of leaf but only by species. Finally, from the formal perspective, I recommend revising lines 82-84, rearranging the sentence in order to avoid redundancies (e.g., "sampling" and "samples"), correcting the plurals (e.g., each sample instead of "each samples") and not omitting the verbs (in the phrase " and in air-tight bags at -80℃" verb is missing).

L 85: please, add a comma after “Chemical Composition”. Thanks.

L 86-87: Tables 1 and 3 are both shown. Why is "or" used, as if there were either one or the other? Apart from this, Table 3 summarizes results and certainly not methodologies or characteristics and material data. Therefore table 3 should not be mentioned in this section. Thanks.

L 89: authors are asked to report the term in vitro consistently throughout the text, or by uniquely choosing italics or not. Thanks.

L 90: please specify the methods used to analyze the chemical composition [17-19] in the same way as done elsewhere (e.g., lines 91-94). In addition, each analytical assay performed should be adequately detailed, as well as how many replicates were performed for each analysis (I read three replies only at the foot of table 3). Thanks.

L 101: please, correctly write "Animals". Thanks.

L 166: Could the authors specify the acronym “ILM”? Thanks.

L 173 (and along with the text): according to the journal template, the p-value should be reported in lowercase and italics. Thanks.

L 191-192: Mention has already been made to materials and methods for assessing digestibility and pH. I suggest avoiding unnecessary redundancies, thanks.

L 101-102, 212-213, 224-225, 245-246, 251-252: see the comment for L 173. Thanks.

L 254-255: I apologize for my pedanticism, but the results must be interpreted and discussed in this section and not recalled again. Thanks.

Author Response

Response to Reviewer 3 Comments

animals-1853702

GENERAL REMARKS

Dear authors,

I have evaluated your manuscript identified as animals-1853702. The manuscript addresses the potential of bamboo foliage as an alternative to hay in the feeding of ruminants. Considering that the literature on this subject is scarce, your contribution could be greatly appreciated. That said, I found some inaccuracies as well as I identified some room for improvement, for some parts. My main criticisms concern the materials and methods, specifically paragraphs 2.1 and 2.2, in which the description of the sampling and analysis methods should be better detailed and described, in my opinion. Even the introduction has some critical issues and I think it can also be revised to make the work more captivating and more visible for readers. On the other hand, except for a few minor details, both the results and the discussions are well organized and detailed in my opinion. The conclusions are fine. My specific comments are listed below, point by point. I hope my contribution will improve the quality of the manuscript. Good work.

SPECIFIC COMMENTS

Point 1: L 17-18: the purposes that the authors highlighted (i.e., to mitigate enteric emissions) are true for all ruminants, rather than just cattle. Since the manuscript is aimed at an exploratory in vitro study, it is my opinion that it is more useful to generalize by referring to all ruminants. Thanks.

Response 1: Thank you for reviewer’s comment! We have changed “cattle” to “ruminants”. (Line 18)

Point 2: L 23: please, replace “livestock” with “ruminants”. Thanks.

Response 2: Thank you for reviewer’s comment! We have changed “livestock” to “ruminants”. (Line 23)

Point 3: L 24: please, replace “suppression” with “reduction”. Thanks.

Response 3: Thank you for reviewer’s comment! We have changed “suppression” to “reduction”. (Line 24)

Point 4: L 25: I suggest recasting the sentence avoiding the syntagma “in in vivo”. Thanks.

Response 4: Thank you for reviewer’s comment! We have changed “in in vivo” to “in vivo”. (Line 26)

Point 5: L 29-30: in my opinion, the two bamboo species tested in the study should be already mentioned in this part of the text. Thanks.

Response 5: Thank you for reviewer’s comment! We have mentioned two bamboo species in this part of the text. (Line 30)

Point 6: L 43: in my opinion, "waste recycling" or "circular economy" can be included in the keywords. Thanks

Response 6: Thank you for reviewer’s comment! We have added the keyword “waste recycling”. (Line 44)

Point 7: L 47-48: authors are requested to check punctuation. In addition, the phrase "Greenhouse gas emissions from the livestock industry are estimated to constitute 14.5 — 19% of global emissions" should be supported by adequate references. Thanks.

Response 7: Thank you for reviewer’s comment! We have corrected the punctuation. (Line 49)

Point 8: L 52: please, delete the comma between “economy” “and the environment”. Moreover, the article on the environment is superfluous Thanks.

Response 8: Thank you for reviewer’s comment! We have deleted environment. (Line 54)

Point 9: L 53-57: recovery and valorization of agro-industrial residues are currently indicated as key factors for the development of the circular economy models and to promote the environmental sustainability of production systems. International scientific opinion is very interested in these issues, which the authors have just mentioned in the simple summary (i.e., 21-22). In my opinion, however, it should be stressed more throughout the text and, particularly, in the introduction. I suppose that this view is valid because the cultivation of bamboo is mainly aimed at the production of lignocellulosic biomass, in which the leaves constitute a waste of the supply chain. So, I would suggest to the authors, albeit briefly, mention the referenced supply chains and the foliage as their by-products and the possibility of reuse of residues between agricultural supply chains. In this regard, I suggest using manuscript doi:10.3390/ani9110918 as a template (especially the introduction), which I strongly recommend using as a reference. Thanks

Response 9: Thank you for reviewer’s comment! We added the sentence by referring to the references commended by the reviewer and other references to refer to the reviewers' comments. (Line 58-61)

“Bamboo is used for construction and pulp production, which results in significant amounts of bamboo leaves as a by-product. Therefore, the agro-industrial residues are useful as animal feed or additives to promote environmental sustainability of production systems.”

Reference : Véras, R.M.L.; Gois, G.C.; Nascimento, D.B.; Magalhães, A.L.R.; Teodoro, A.L.; Pinto, C.S.; Oliveira, L.P.; Campos, F.S.; Andrade, A.P.; Lima, I.E. Potential Alternative Feed Sources for Ruminant Feeding from the Biodiesel Production Chain By-Products. S. Afr. J. Anim. Sci. 2020, 50, 69–77.

Serrapica, F.; Masucci, F.; Raffrenato, E.; Sannino, M.; Vastolo, A.; Barone, C.M.A.; Di Francia, A. High Fiber Cakes from Mediterranean Multipurpose Oilseeds as Protein Sources for Ruminants. Animals 2019, 9, 918.

Point 10: L 55: the area covered with bamboo that the authors report refers only to China or to another geographical scale. The authors are requested to clarify, thanks.

Response 10: Thank you for reviewer’s comment! We have clarified the our references from bamboo. (Line 58)

Point 11: L 57: the use of the term "previous studies" implies that reference is made to the studies referred to. However, no indication is given in this regard. Authors are requested to rearrange the sentence by noting the studies to which they refer. Thanks.

Response 11: Thank you for reviewer’s comment! We have added references. (Line 64)

Point 12: L 69-71: on what basis did the authors formulate their hypothesis? From the literature data reported above the hypothesis cannot be guessed. I think the authors should better formulate the sentence, since what is reported as a hypothesis seems to coincide more with an expected result. Alternatively, greater clarification of the state of the art could help to understand the hypothesis formulated by the authors. Thanks.

Response 12: Thank you for reviewer’s comment! We have added references to guess our hypothesis. (Line 69-71)

“Recently, many studies have shown that the biologically active compounds of by-products have the potential to mitigate methane production by modulating ruminal microbial population”

Reference : Huang, H.; Szumacher-Strabel, M.; Patra, A.K.; Ślusarczyk, S.; Lechniak, D.; Vazirigohar, M.; Varadyova, Z.; Kozłowska, M.; Cieślak, A. Chemical and Phytochemical Composition, in Vitro Ruminal Fermentation, Methane Production, and Nutrient Degradability of Fresh and Ensiled Paulownia Hybrid Leaves. Anim. Feed Sci. Technol. 2021, 279, 115038.

Lee, S.J.; Kim, H.S.; Eom, J.S.; Choi, Y.Y.; Jo, S.U.; Chu, G.M.; Lee, Y.; Seo, J.; Kim, K.H.; Lee, S.S. Effects of Olive (Olea Europaea L.) Leaves with Antioxidant and Antimicrobial Activities on In Vitro Ruminal Fermentation and Methane Emission. Animals 2021, 11, 2008.

Point 13: L 71-74: perhaps it would be better if the authors avoided generalizing. In fact, more than just bamboo leaves, the authors have analyzed leaves of two specific species, which in my opinion should be clearly mentioned. Thanks.

Response 13: Thank you for reviewer’s comment! We have added leaves of two specific species to avoid generalization. (Line 79)

Point 14: L 80-84: in my opinion, there are several inconsistencies in this sentence. First, it is unclear whether the leaves of eight plants were used for each species tested or whether the eight plants refer to both species (4 plants/species). It is also not clear whether the "young and mature" leaves (the term "young" is not very technical and should be replaced with immature ") were harvested progressively over the indicated time (May and June) and if so, in what sequence. Given the differences in chemical composition between the leaves of the two species, these aspects are not marginal. I'm not sure if the leaves collected were grouped by type of leaf within each species or simply by species of origin. The statement "were harvested and pooled for each species of leaves" creates perplexity in this regard. Considering what is reported below [materials and methods (paragraphs 2.3 and 2.4) and results] I have not noticed any analysis differentiated by the type of leaf but only by species.

Response 14: Thank you for reviewer’s comment!

First. We used "type" to mean species. However, it was deleted because it was unclear as the reviewer comment. (Line 87)

Second. We have changed “young” to “immature”. (Line 87) and added in what sequence they were harvested. (Line 89-90)

We have added “samples classified by species of origin” for clarity. (Line 90)

Point 15: Finally, from the formal perspective, I recommend revising lines 82-84, rearranging the sentence in order to avoid redundancies (e.g., "sampling" and "samples"), correcting the plurals (e.g., each sample instead of "each samples") and not omitting the verbs (in the phrase " and in air-tight bags at -80℃" verb is missing).

Response 15: Thank you for reviewer’s comment! We have revised the Line (avoid redundancies, correcting the plurals and not omitting the verbs). (Line 90-92)

Point 16: L 85: please, add a comma after “Chemical Composition”. Thanks.

Response 16: Thank you for reviewer’s comment! We have added a comma after “Chemical Composition”. (Line 93)

Point 17: L 86-87: Tables 1 and 3 are both shown. Why is "or" used, as if there were either one or the other? Apart from this, Table 3 summarizes results and certainly not methodologies or characteristics and material data. Therefore table 3 should not be mentioned in this section. Thanks.

Response 17: Thank you for reviewer’s comment! We used "or" because we did not evaluate the antioxidant capacity of the substrate. Therefore, we have changed this section according to the reviewer’s suggestion. (Line 94-95)

Point 18: L 89: authors are asked to report the term in vitro consistently throughout the text, or by uniquely choosing italics or not. Thanks.

Response 18: Thank you for reviewer’s comment! We have changed the term in vitro to italics to be consistent throughout the text.

Point 19: L 90: please specify the methods used to analyze the chemical composition [17-19] in the same way as done elsewhere (e.g., lines 91-94). In addition, each analytical assay performed should be adequately detailed, as well as how many replicates were performed for each analysis (I read three replies only at the foot of table 3). Thanks.

Response 19: Thank you for reviewer’s comment! We have added methods used to analyze the chemical composition (Line 98) and how many replicates of all analyses (Line 106-108).

Point 20: L 101: please, correctly write "Animals". Thanks.

Response 20: Thank you for reviewer’s comment! We have changed “Anmials” to “Animals”. (Line 112).

Point 21: L 166: Could the authors specify the acronym “ILM”? Thanks.

Response 21: Thank you for reviewer’s comment! We have added acronym “IML”. (Line 180)

Point 22: L 173 (and along with the text): according to the journal template, the p-value should be reported in lowercase and italics. Thanks.

Response 22: Thank you for reviewer’s comment! We have changed the p-value to lowercase and italics according to the journal template.

Point 23: L 191-192: Mention has already been made to materials and methods for assessing digestibility and pH. I suggest avoiding unnecessary redundancies, thanks.

Response 23: Thank you for reviewer’s comment! We have avoided the unnecessary redundancies (method for digestibility and pH). (Line 206-207)

Point 24: L 101-102, 212-213, 224-225, 245-246, 251-252: see the comment for L 173. Thanks.

Response 24: Thank you for reviewer’s comment! We have changed the p-value to lowercase and italics according to the journal template.

Point 25: L 254-255: I apologize for my pedanticism, but the results must be interpreted and discussed in this section and not recalled again. Thanks.

Response 25: Thank you for reviewer’s comment! We have deleted the results. (Line 271)

Thank you for investing your time and reviewing the manuscript.

Round 2

Reviewer 2 Report

The authors have corrected some aspects of this article. However, there are still several things, indicated during the first correction, that have not been corrected. In particular, there are some considerations in the results section that definitely need to be revised. In addition, some indications in the discussions were not made. I therefore submit the previous version of my corrections

Author Response

Response to Reviewer 2 Comments

The authors have corrected some aspects of this article. However, there are still several things, indicated during the first correction, that have not been corrected. In particular, there are some considerations in the results section that definitely need to be revised. In addition, some indications in the discussions were not made. I therefore submit the previous version of my corrections

Point 1: What do you mean with with "pulp production"? Please, specify

Response 1: Thank you for reviewer’s comment! We have changed “pulp production” to “paper pulp production”. (Line 21)

Point 2: Please spcify which part of bamboo

Response 2: Thank you for reviewer’s comment! We have added specify which part of bamboo. (Line 23)

Point 3: Lines 31-32: Please specifiy that you evaluated two different species of bamboo

Response 3: Thank you for reviewer’s comment! We have changed “bamboo leaves” to “Phyllostachys nigra var. henonis (PHN) and Sasa borealis”. (Line 30)

Point 4: and consequently instead of subsequently

Response 4: Thank you for reviewer’s comment! We have changed “subsequently” to “consequently”. (Line 29)

Point 5: Da quello che ho capito le foglie sono un residuo della lavorazione del bamboo. Sarebbe interessante, implementare l'introduzione aggiunge qualcosa riguardo la riduzione degli sprechi, e l'utilizzo dei residui in alimentazione animale.

Considering the following article:

A review on the use of agro-industrial CO-products in animals’ diets. Vastolo, A., Calabrò, S., Cutrignelli, M.I.

Response 5: Thank you for reviewer’s comment! We added the sentence. (Line 56-59)

“Bamboo is used for construction and pulp production, which results in significant amounts of bamboo leaves as a by-product. Therefore, the agro-industrial residues are useful as animal feed or additives to promote environmental sustainability of production systems.”

Reference : Véras, R.M.L.; Gois, G.C.; Nascimento, D.B.; Magalhães, A.L.R.; Teodoro, A.L.; Pinto, C.S.; Oliveira, L.P.; Campos, F.S.; Andrade, A.P.; Lima, I.E. Potential Alternative Feed Sources for Ruminant Feeding from the Biodiesel Production Chain By-Products. S. Afr. J. Anim. Sci. 2020, 50, 69–77.

Serrapica, F.; Masucci, F.; Raffrenato, E.; Sannino, M.; Vastolo, A.; Barone, C.M.A.; Di Francia, A. High Fiber Cakes from Mediterranean Multipurpose Oilseeds as Protein Sources for Ruminants. Animals 2019, 9, 918.

Point 6: Please add the following reference:

UHPLC-ESI-QqTOF Analysis and In Vitro Rumen Fermentation for Exploiting Fagus sylvatica Leaf in Ruminant Diet. Formato, M., Piccolella, S., Zidorn, C., (...), Cutrignelli, M.I., Pacifico

Response 6: Thank you for reviewer’s comment! We added the reference. (Line 64)

Point 7: This sentece is refered more to the results paragraph than materials and methods. Please delate

Response 7: Thank you for reviewer’s comment! We have changed the sentence to “Chemical composition of substrates used on in vitro experiments is shown in Table 1”. (Line 91)

Point 8: You modified the method of Brand-Williams, in which way?

Response 8: Thank you for reviewer’s comment! We have modified method as given in Seleshe, Semeneh, et al's article. (Line 99)

Reference :“Seleshe, Semeneh, et al. "Evaluation of antioxidant and antimicrobial activities of ethanol extracts of three kinds of strawberries." Preventive Nutrition and Food Science 22.3 (2017): 203.”

Point 9: Considering that you are reported the chimical composition of an hay, could be important reporting the lignin (ADL) value

Response 9: Reviewer has raised an important point. However, we thought that NDF and ANF analysis could produce results. In the next experiment, we will analyze the lignin content and fulfill at feed ingredient information. We hope this response will be acceptable to reviewer. (Line 105)

Point 10: Please, delete substrate and reported control diet

Response 10: Thank you for reviewer’s comment! We have changed “substrate” to “control diet”. (Line 105)

Point 11: This sentence is too long, please reduce. Furthermore, it is not clear how the substrates were incubated. Different concentrations of bamboo leaves were included together with 300 mg substrate (Timothy hay). I do not understand why only hay was used as a control substrate and not a standard cattle diet consisting of forages and concentrates. Or evaluate the inclusion of bamboo leaves (fodder) together with a concentrate.

Response 12: Thank you for reviewer’s comment!

We have changed the sentence.

We have added the actual trial design from table (Line 128) and why timothy was used as a control substrate (Line 122-123).

The actual trial design was:

5 incubation times × 7 treatments × 4 repetitions × 2 runs

“Timothy was used as a control because it has similar chemical composition to bamboo leaves and is a commonly used feed for ruminants.”

Why did you include 300 mg of Timothy hay? Maybe it's to low. Do you have any reference that explain this dose?

We have added the sentences from explain this dose. (Line 123-124)

“The amount of timothy was chosen based on a previous study.”

Reference :“Lee, Shin Ja, et al. "Effects of Olive (Olea europaea L.) Leaves with Antioxidant and Antimicrobial Activities on In Vitro Ruminal Fermentation and Methane Emission." Animals 11.7 (2021): 2008.”

Please clarify

Point 13: I don't understand, did you asses the same parameter with two different methods?

Response 13: Thank you for reviewer’s comment! We have changed and deleted the sentences.

“The apparent DM digestibility was determined following the nylon bag digestion method. Briefly, the residues were washed in a water bath and dried in a dry oven at 105 ℃ for 24 h.” (Line 145-149)

Point 14: The incubation lasted also 72h, Why did you collecte the rumen liquor of 48h?

Response 14: Reviewer has raised an important point. There are two reasons for displaying and discussing only 48 h incubations of microbial assays. First, when we conducted in vitro batch culture experiments, the most differences observed at 48 h incubation(gas and vfa profiles). Second, microbial analysis in the cultures could be conducted only in a limited number of samples due to economic and technical constraints. We hope this reply will be acceptable to the reviewer. (Line 151)

Point 15: Please clarify, seems there is a repetion in this period

Response 15: Thank you for reviewer’s comment! We have added the sentence. (Line 174)

“All the reaction were performed in triplicate.”

Point 16: NDICP

Response 16: Thank you for reviewer’s comment! We have changed “NDFIP” to “NDICP”. (Line 193)

Point 17: If you have done a statistical analysis of the chemical composition and other parameters, you MUST report either the letters to indicate the differences or the p<value. Otherwise, it's a atable referes to materials and methods’

Response 18: Thank you for reviewer’s comment! We think statistical analysis comparing the chemical composition and other parameters of two species of bamboo leaves will be difficult, because there are no controls. We refer to references that include chemical composition analysis without statistical analysis in the results. We hope this reply will be acceptable to the reviewer. (Line 199)

Reference :“Soliva, C. R., et al. "Feeding value of whole and extracted Moringa oleifera leaves for ruminants and their effects on ruminal fermentation in vitro." Animal feed science and technology 118.1-2 (2005): 47-62.”

“Pal, K., et al. "Evaluation of several tropical tree leaves for methane production potential, degradability and rumen fermentation in vitro." Livestock Science 180 (2015): 98-105.”

Point 18: Table 4

Response 18: Thank you for reviewer’s comment! We have changed “Table 3” to “Table 4”. (Line 204)

Point 19: I don't agree, at 24h all values are the highest except for 6.59 that is the lowest. Maybe could be better to report this last

Response 19: Thank you for reviewer’s comment! We have deleted the 24 h. (Line 207)

Point 20: Do you have a control, so maybe could be better perform a Dunnett test, Post-Hoc test which compare control to experimental thesis

Response 20: Thank you for reviewer’s comment! In our opinion, the polynomial contrast was used because there is a variable that is the dose response level, and the tukey test, which is a post hoc test to compare the overall results, was used. As the reviewer said, it would be good to use the Dunnett test in an experiment using various additive. We hope this reply will be acceptable to the reviewer. (Line 211)

Point 21: Why the ph has been asses at 12-24-48h and not for example at 6-24-72h?

Response 21: Reviewer has raised an important point. There are one reason why only displayed and discussed for 12, 24 and 48 h incubation. First, when we conducted in vitro batch culture experiments, the differences observed at 12, 24, and 48 h incubation. However, we have to check pH, DM digestibility, gas profiles, and VFA profiles up to 48 h incubation is because we thought that this is an essential factors in determining as a feed ingredients. We hope this reply will be acceptable to the reviewer. (Line 211)

Point 22: No significant differences between the diets?

These one are significant and PHN not, again seem strange. Check the standard devation

that's impossible

Response 22: Thank you for reviewer’s comment! We did the statistical analysis again, but the results were the same. (Lien 211)

Point 23: In Table 5, total gas...

Response 23: Thank you for reviewer’s comment! We have added “In Table 5,”.(Line 217)

Point 24: 0.01

Response 24: Thank you for reviewer’s comment! We have changed “0.05” to “0.01”. (Line 217)

Point 25: You campare all the diets, How you can say that?. It's better if you say that the CON was the highest

Response 25: Thank you for reviewer’s comment! We have changed the sentence.

“In Table 5, total gas production decreased linearly (p < 0.01) with increasing doses of bamboo leaves supplementation, and CON group was the highest (p < 0.05).” (Line 217-218)

Point 26: The are interesting results also at 12h

Response 26: Thank you for reviewer’s comment! We have added 12 h. (Line 220)

Point 27: Why did you not report the results obteined at 72h?

Response 27: Reviewer has raised an important point. There are one reason why only displayed and discussed for 12, 24 and 48 h incubation. First, when we conducted in vitro batch culture experiments, the differences observed at 12, 24, and 48 h incubation. However, we have to check pH, DM digestibility, gas profiles, and VFA profiles up to 48 h incubation is because we thought that this is an essential factors in determining as a feed ingredients. We hope this reply will be acceptable to the reviewer. (Line 223)

Point 28: and linearly

Response 28: Thank you for reviewer’s comment! We have added “linearly”. (Line 230)

Point 29: Also at 24h

Response 29: Thank you for reviewer’s comment! We have added “24 h”. (Line 233)

Point 30: I suggest to report these results in table

As figure 1

Response 30: Thank you for reviewer’s comment! We adjusted the size of Figures 1 and 2 to make it more visible. However, if the reviewer says, "It needs to be changed," we will revise it. We hope this reply will be acceptable to the reviewer. (Line 258 and 264)

Point 31: Please add some discussion on the degestibility of DM, which increases and the chemical composition.  Both are interesting results, you should discuss them

Response 33: Thank you for reviewer’s comment! We have added the sentence. (Line 295-297)

“The DM digestibility by bamboo leaves supplementation was approximately 4.9—11.2% improved than CON group, which may be due to influenced by plant secondary metabolites of plant.”

Reference :Hussain, I.; Cheeke, P.R. Effect of Dietary Yucca Schidigera Extract on Rumen and Blood Profiles of Steers Fed Concentrate-or Roughage-Based Diets. Anim. Feed Sci. Technol. 1995, 51, 231–242.

Point 32: But there was a variance of ph among the diet. I did not understand this sentence

Response 32: Thank you for reviewer’s comment! We have changed the sentence. (Line 287-288)

“Here, bamboo leaves supplementation induced a shift in the VFA pattern and affected the pH.”

Point 33: I don't agree, there is a significant reduction at 12h.

Response 33: Thank you for reviewer’s comment! We have changed the sentence. (Line 298-301)

“Methane production was not affected by bamboo leaves supplementation after 24 h incubation, but 12 and 48 h reduced approximately 12.7—21.9% more than CON group, without compromising total VFA production.”

Reviewer 3 Report

Dear authors,
I have evaluated the revised version of your manuscript identified as animals-1853702. Given the changes made to the previous version, I have no other suggestions. As a minor detail, I noticed a typo (I suppose!) on line 336 of the new version. Apart from that, I believe the current version of the manuscript deserves to be published without any other changes. Best regards and congratulations

Author Response

Response to Reviewer 3 Comments

Dear authors,
I have evaluated the revised version of your manuscript identified as animals-1853702. Given the changes made to the previous version, I have no other suggestions. As a minor detail, I noticed a typo (I suppose!) on line 336 of the new version. Apart from that, I believe the current version of the manuscript deserves to be published without any other changes. Best regards and congratulations

Thank you for reviewer’s comment! We have changed the typo. Thank you for investing your time and reviewing the manuscript.
